# Polysarcosine-Functionalized mRNA Lipid Nanoparticles Tailored for Immunotherapy

**DOI:** 10.3390/pharmaceutics15082068

**Published:** 2023-08-01

**Authors:** Christoph Wilhelmy, Isabell Sofia Keil, Lukas Uebbing, Martin A. Schroer, Daniel Franke, Thomas Nawroth, Matthias Barz, Ugur Sahin, Heinrich Haas, Mustafa Diken, Peter Langguth

**Affiliations:** 1Department of Biopharmaceutics and Pharmaceutical Technology, Johannes Gutenberg University Mainz, 55128 Mainz, Germany; cwilhelmy@uni-mainz.de (C.W.);; 2TRON—Translational Oncology at the University Medical Center of Johannes Gutenberg University gGmbH, 55131 Mainz, Germany; isabell.keil@tron-mainz.de; 3European Molecular Biology Laboratory (EMBL) Hamburg Outstation, c/o DESY, 22607 Hamburg, Germany; 4Nanoparticle Process Technology (NPPT), Faculty of Engineering, University of Duisburg-Essen, 47057 Duisburg, Germany; 5BIOSAXS GmbH, 22607 Hamburg, Germany; 6LACDR—Leiden Academic Centre for Drug Research, Leiden University, 2333 Leiden, The Netherlands; 7Department of Dermatology, University Medical Center, Johannes Gutenberg University Mainz, 55131 Mainz, Germany; 8Department of Immunology, University Medical Center, Johannes Gutenberg University Mainz, 55131 Mainz, Germany; 9BioNTech SE, 55131 Mainz, Germany

**Keywords:** lipid nanoparticles, LNPs, mRNA, small-angle X-ray scattering, polysarcosine, flow cytometry, immunotherapy, cancer, vaccine

## Abstract

Lipid nanoparticles (LNPs) have gained great attention as carriers for mRNA-based therapeutics, finding applications in various indications, extending beyond their recent use in vaccines for infectious diseases. However, many aspects of LNP structure and their effects on efficacy are not well characterized. To further exploit the potential of mRNA therapeutics, better control of the relationship between LNP formulation composition with internal structure and transfection efficiency in vitro is necessary. We compared two well-established ionizable lipids, namely DODMA and MC3, in combination with two helper lipids, DOPE and DOPC, and two polymer-grafted lipids, either with polysarcosine (pSar) or polyethylene glycol (PEG). In addition to standard physicochemical characterization (size, zeta potential, RNA accessibility), small-angle X-ray scattering (SAXS) was used to analyze the structure of the LNPs. To assess biological activity, we performed transfection and cell-binding assays in human peripheral blood mononuclear cells (hPBMCs) using Thy1.1 reporter mRNA and Cy5-labeled mRNA, respectively. With the SAXS measurements, we were able to clearly reveal the effects of substituting the ionizable and helper lipid on the internal structure of the LNPs. In contrast, pSar as stealth moieties affected the LNPs in a different manner, by changing the surface morphology towards higher roughness. pSar LNPs were generally more active, where the highest transfection efficiency was achieved with the LNP formulation composition of MC3/DOPE/pSar. Our study highlights the utility of pSar for improved mRNA LNP products and the importance of pSar as a novel stealth moiety enhancing efficiency in future LNP formulation development. SAXS can provide valuable information for the rational development of such novel formulations by elucidating structural features in different LNP compositions.

## 1. Introduction

Although lipid-based nano-sized drug delivery systems have been available for several decades, advancements in the last decade have significantly improved their potential to deliver nucleic acids such as DNA, siRNA and most recently mRNA [1,2,3,4]. In particular, mRNA lipoplexes (LPXs) and lipid nanoparticles (LNPs) have shown promising results in preclinical and clinical studies, with therapeutic applications such as cancer immunotherapy [5,6,7]. LNPs were most recently utilized for vaccinations against the SARS-CoV-2 virus, the cause of COVID-19 [8,9]. As a consequence of this success, extensive development activities at all stages, ranging from early research to the clinical stage for LNP-based mRNA products, have emerged. The LNP formulations need to fulfill the following properties for optimal mRNA delivery: encapsulation and protection of nucleic acid from degradation by ubiquitous nucleases, adequate long circulation in the bloodstream to reach their destination and successful delivery of RNA to target cells or organs of interest [4,10].

In general, LNP formulations consist of an ionizable-cationic lipid, cholesterol, a helper lipid or phospholipid and a polyethylene glycol (PEG) lipid as stealth moiety [11]. While most of these LNP drug products share this basic concept, the actual composition can vary considerably between different formulations. However, the structural effects of these variations remain relatively unexplored.

A key component of the LNP is given by the ionizable lipid, which is considered essential for efficient endosomal escape. The increase of positive charge density induced by the drop in pH upon acidification of the late endosome is thought to promote the binding and rupture of the endosomal membrane [12,13,14,15]. Jayaraman et al. showed that the ionizable lipid should have a pK_a_ of ~6.5 for the highest activity of hepatic siRNA delivery [16]. Recently, for the first time, we were able to elucidate the pH-induced changes in the bilayer structure in LPXs comprising ionizable lipids in situ by using small-angle X-ray scattering (SAXS) [17], where we could reveal clear differences as a function of lipid structure and the composition of the systems.

As helper lipids, typically phospholipids comprising either a phosphatidyl ethanolamine (-PE) or phosphatidyl choline (-PC) head group are used, where lipids with saturated or unsaturated hydrocarbon chains may be selected. The helper lipids may have an important influence on the activity and the targeting selectivity of the LNPs. For example, it has been shown that by using dioleyl phosphatidylethanolamine (DOPE) in LPXs as a helper lipid, the mRNA expression in the liver could be reduced to very low levels [18,19]. For many future applications of mRNA, therapeutics such as extrahepatic targeting, or more generally speaking, organ-selective targeting, is still an unmet need, since classical LNPs typically result in the highest expression in the liver. The selection of appropriate helper lipids and lipid compositions may allow the better adjustment of targeting selectivity according to the therapeutic requirements [20].

The use of PEG-functionalized lipids (PEGylated lipids) as an excipient for drug and mRNA delivery has been widely adopted in various nanoparticulate systems, including liposomes, proteins and LNPs. In addition to modulating circulation in the bloodstream, PEGylation is necessary for LNP engineering to avoid aggregation during the mixing of the RNA with the lipid solution. Besides the potential problems in safety and tolerability of the ionizable cationic lipids [21,22], recent studies have also raised concerns about the use of PEG in drug delivery systems, with PEG-lipids being hypothesized to be the cause of several adverse reaction phenomena, such as complement-activation-related pseudoallergy (CARPA) causing hypersensitivities. One further problem is its immunogenicity, which can trigger the production of undesired anti-PEG-antibodies, potentially facilitating the accelerated blood clearance (ABC) phenomenon following repeated administrations [23,24,25]. Furthermore, it was demonstrated that anti-PEG-antibodies are compromising the bilayer integrity, which can induce premature drug release [26]. The exposure with PEG-containing products, as is often the case for cosmetics and household products, increases the probability of anti-PEG-antibodies in the population [24]. These concerns led to the research for alternative excipients replacing PEG as stealth moiety. In this context, the use of polysarcosine (pSar) has shown promising results as an excipient providing stealth properties [27,28,29,30], together with enhanced nanoparticle stability and reduced immunogenicity [31,32]. For mRNA LNPs, we have previously demonstrated that pSar lipids are a versatile tool for LNP engineering, where equivalent or better activity together with reduced immunogenicity can be obtained [33].

SAXS is a powerful tool to gain information on the structure and lamellarity and internal organization inside lipid nanoparticles, which is also considered a quality parameter in FDA guidance for liposomal drug formulations [34]. Previous studies in our group using SAXS elucidated the importance of structural features for mRNA delivery systems, including polymer and lipid-based formulations [17,35,36,37]. 

In this study, we investigated selected fundamental formulation parameters for mRNA LNP manufacturing by characterizing the physicochemical characteristics and activity in vitro. We used an improved, single-step manufacturing process for the LNPs which does not require dialysis or tangential flow filtration to obtain the injectable product. We investigated permutations from combinations between two different well-established ionizable lipids together with phospholipids comprising either a -PE or a -PC headgroup and two different stealth moieties. All systems comprised the same amount of cholesterol. The ionizable lipids were 1,2-dioleyloxy-3-dimethylaminopropane (DODMA), the ionizable variety of one of the longest-known cationic lipids [38], and DLin-MC3-DMA (MC3), which is being used in the marketed product Onpattro [2]. As helper lipids, we compared 1,2-dioleoyl-sn-glycero-3-ethanolamine (DOPE) and 2-Dioleoyl-sn-glycero-3-phosphocholine (DOPC), which share an identical backbone. The stealth moieties were PEG, the gold standard for RNA delivery in the form of C16-PEG-Ceramide lipid, and pSar, a relatively new stealth moiety linked to a C12 bisalkyl amine [39]. 

Our results show that SAXS enabled the sensitive determination of the influence of the respective lipids on the LNP structure. This accurate insight into the molecular organization of the particles allowed the derivation of refined correlations with their potency, beyond the usually described pK_a_ value of the ionizable lipid. pSar could be successfully applied for particle manufacturing, resulting in a bit larger LNP sizes, but with low influence on internal structure. pSar formulations showed better in vitro activity compared to those made with PEG. This highlights the potential of pSar as an improved stealth moiety and promising alternative to PEG within mRNA LNP formulations. 

## 2. Materials and Methods

### 2.1. Materials

1,2-Dioleoyl-sn-glycero-3-phosphocholine (DOPC) and 1,2-dioleoyl-sn-glycero-3-ethanolamine (DOPE) were manufactured by Lipoid GmbH (Ludwigshafen, Germany). 1,2-dioleyloxy-3-dimethylaminopropane (DODMA), N-palmitoyl-sphingosine-1-{succinyl[methoxy(polyethylene glycol)2000]} (C16-PEG2000-Ceramide) and Cholesterol were manufactured by Avanti Polar Lipids (Alabaster, AL, USA). (6Z,9Z,28Z,31Z)-heptatriacont-6,9,28,31-tetraene-19-yl-4-(dimethylamino) butanoate (DLin-MC3-DMA) was purchased from MedChemExpress (Monmouth Junction, NJ, USA). pSar BA12-50 (didodecyl amine initiated polysarcosine with a pSar chain length of 44, as determined by 1H-NMR) was synthesized as previously described [39]. Three types of synthetic mRNA were used, synthetized by BioNTech SE (Mainz, Germany), using internal protocols [40]. Non-coding R159 mRNA consisting of 1900 nucleotides was used for physicochemical characterization. Thy1.1 (CD90.1) encoding reporter mRNA consisting of 1064 nucleotides was used for in vitro transfection assay. Cy5-labeled Luc encoding reporter mRNA consisting of 2135 nucleotides was used for in vitro cell-binding studies. Uridine-Triphosphate (UTP) labeled with Cy5 was purchased from Jena Bioscience (Jena, Germany). 6-(p-toluidino)-2-naphthalenesulfonic acid (TNS), KH_2_PO_4_ and Na_2_HPO_4_ were obtained from Sigma-Aldrich (St. Louis, MO, USA), while Ampuwa was purchased from Fresenius Kabi Deutschland GmbH (Bad Homburg vor der Höhe, Germany). Dimethyl sulfoxide (DMSO) was purchased from VWR International GmbH (Darmstadt, Germany). Absolute ethanol (200 proof) and Quant-it™ RiboGreen RNA reagent was purchased from Fisher Scientific (Schwerte, Germany) and glycylglycine was obtained from Carl Roth (Karlsruhe, Germany). For in vitro transfection and cell-binding assay, thawed hPBMCs were resuspended in RPMI medium1640 (1×) + GlutaMAX-I, Sodium Pyruvate 100 mM (100×), MEM NEAA (100×), 2-Mercaptoethanol (50 mM), Pen Strep, DPBS (1×) 0.5 m EDTA pH 8.0 and Pooled Human Serum (PHS; heat inactivated), purchased from Life technologies (San Diego, CA, USA). Cells were washed during in vitro cell-binding assay in DPBS (1×), 5% FBS (heat inactivated) and 5 mM EDTA (500 mM), purchased from Life technologies (San Diego, CA, USA).

### 2.2. RNA Constructs and In Vitro Transcription

Plasmid templates for the in vitro transcription of protein-encoding RNA were used as reporters. The Thy1.1 vector encodes the murine Thy1.1 protein, a highly conserved membrane glycoprotein. The Luc vector encodes for luciferase protein. Thy1.1 and Luc RNA were synthesized using 1-methyl-pseudouridine (N1-methylpseudouridine-5′-triphosphate, m1ΨTP, TriLink Biotechnologies, San Diego, CA, USA) [41] and double-stranded mRNA (dsmRNA) contaminants were removed via cellulose purification, as described [42]. The labeling of Luc RNA with fluorescent Cy5-UTP was performed according to the manufacturer’s instructions (BioNTech SE, Mainz, Germany). During the in vitro transcription of Luc RNA, 6% of total UTP was replaced with Cy5-labeled UTP.

### 2.3. LNP Preparation

LNPs were prepared by using the ethanol injection method as used for LPXs [17] in a modified variation. The mRNA was diluted to the predetermined concentration with 10 mM aqueous glycylglycine solution (pH 5.7 ± 0.1) and the ethanolic lipid stock solutions were pre-mixed to obtain the predetermined molar ratios. The aqueous mRNA buffer was pipetted onto the lipid mix and the sample was then immediately vortexed for 10 s. This single-step protocol allows the manufacturing of mRNA LNPs in a directly applicable buffer without further modification. All LNPs were prepared at a molar ratio of ionizable lipid to mRNA (N/P ratio) of 5:1. Molar ratios were calculated as 100 mol% = ∑ mol% (helper lipid, ionizable lipid, cholesterol, stealth moiety). The LNP composition can be seen in Table 1.

### 2.4. Dynamic Light Scattering and Zeta Potential Measurements

The samples were diluted to an appropriate concentration (1 mg/mL for size, 0.1 mg/mL for zeta potential) in glycylglycine buffer and transferred to a Zetasizer Nano ZS (Malvern Panalytical Ltd., Malvern, UK) for size measurements via dynamic light scattering (DLS) as well as zeta potential determination. The measurements were performed as backscattering measurements (scattering angle: 173°) at 25 °C after a 30 s equilibration time.

### 2.5. Accessible mRNA

The incorporation of the mRNA cargo into the LNPs was determined via the commercially available Quant-iT^™^ RiboGreen^®^ assay, as commonly used for this purpose [43]. The fluorescence intensity was measured after the addition of the RiboGreen RNA reagent to the sample solution (*F*_0_) and compared to that after incubation with 0.02% Triton X-100 used to disrupt the LNPs and release the mRNA load (*F_t_*). The disruption of the LNPs after the addition of Triton X-100 was shown via DLS. The inaccessible mRNA rate was calculated as
(1)$$inaccessible\;mRNA=\left(1-\frac{F_{0}}{F_{t}}\right)\cdot 100\;[\%]$$

### 2.6. pK_a_ Fluorescence Assay

A common previously published assay using the fluorescent dye 2-(p-toluidino)-6- naphthalene sulfonic acid (TNS) was performed to determine the apparent formulation pK_a_ [17,44]. Measurements were performed in triplicates on black TC-coated 96-well plates, with each well containing 10 µL sample (at 0.1 mg/mL total lipid), 90 µL buffer (phosphate buffer as proposed by Sörensen from pH 4.5 to pH 9 [45]) and 2 µL of TNS in DMSO (300 µM). The fluorescence was measured from the top on a TECAN infinite 200Pro plate reader (Tecan Group Ltd., Männedorf, Switzerland) at 325 nm excitation and 435 nm emission wavelength.

### 2.7. Small-Angle X-ray Scattering

SAXS measurements were performed at the P12 BioSAXS beamline of the European Molecular Biology Laboratory (EMBL) at PETRA III synchrotron, DESY (Hamburg, Germany) [46]. The samples had at a total lipid concentration of 2.5 mg/mL. The samples and the corresponding buffer solutions were measured at a sample-to-detector distance of 3.0 m (corresponding to a q-range of 0.02–7.37 nm^−1^). The samples and buffers were automatically loaded into an in vacuum flow-through capillary by a robotic sample changer and continuously flowed to reduce radiation damage. The measurements were performed at an X-ray wavelength of 0.124 nm (10 keV energy) and a flux of 5 × 10^12^ ph·s^−1^. A Pilatus 6 M detector (Dectris AG, Baden-Daettwil, Switzerland) was used to collect two-dimensional scattering patterns with an exposure time of 0.095 s. Thirty frames per sample were recorded and only frames without radiation damage were used for averaging. Both the software SASFLOW and the ATSAS software package were used for raw data processing [46,47,48]. The corresponding buffers for all samples were measured as well and subtracted as background signal from the scattering curves. All SAXS profiles are given as a function of the momentum transfer q,
(2)$$q=\frac{4\pi }{\lambda }\cdot \sin\left(\frac{2\theta }{2}\right)$$
where q is defined as the scattering vector, λ is the X-ray wavelength and 2ϴ is the scattering angle.

### 2.8. Data Treatment

Data transformation and analysis were performed using Microsoft Excel (Microsoft, Redmond, DC, USA), QtiPlot 1.0.1 (IONDEV, Bucuresti, Romania) and the ATSAS package (EMBL Hamburg, Hamburg, Germany) [47]. 

As previously reported, SAXS data for the recorded q-range comprise the LNP form factor as well as Bragg reflections from the (period) mRNA packing within the LNPs [33]. A Lorentzian fit functionality in QtiPlot 1.0.1 was utilized for the peak fitting of the Bragg reflections in the SAXS curves, as given in
(3)$$I(q)=I_{0}+\frac{2A}{\pi }\cdot \frac{w}{4\cdot {\left(q-q_{c}\right)}^{2}+w^{2}}$$
with I(q) being the scattering intensity, I_0_ the baseline intensity, A the peak area, w the peak width (FWHM) and q_c_ the peak position. Depending on the scattering data, single or double Lorentzian fits were performed. The peak position can be used to calculate the repeat distance of the scattering moiety (d spacing) for lamellar systems from the Bragg peak using the Bragg equation [49]: (4)$$d=\frac{2\pi }{q_{c}}$$

Additional conclusions can be gained from the peak width w, which gives information about the correlation length inside the ordered arrays. Here, the correlation length scales reciprocally with the peak width, meaning that a narrow peak is an indicator for a long correlation length. A generally accepted model for liquid crystalline structures [50] describes the correlation length ξ, which is defined as the distance at which the positional correlation decays to the value 1/e, as:(5)$$\xi =\frac{2}{w}$$

Additional information can be revealed by analyzing the intensity decay of the whole SAXS profile using the power law as shown in Equation (6), which gives information about fractal dimensionality and the packing compactness of the particles.
(6)$$I\left(q\right)=I_{0}\cdot q^{-x}$$

For particles with smooth surfaces, a steep exponential decay indicated by the so-called Porod slope x with values between 3 and 4 can be observed, while Porod slopes with lower decay in the range of 2–3 are observed for particles with surface fractals [51]. 

### 2.9. In Vitro Thy1.1 Transfection Assay

hPBMCs were isolated via density gradient centrifugation and cryopreserved for further use. For performing in vitro Thy1.1 transfection assay, cryopreserved hPBMCs were thawed at 37 °C in a water bath and resuspended into pre-warmed human DC (hDC) medium (RPMI medium1640 (1×) + GlutaMAX-I containing 5% pooled-human-serum, 1% Sodium Pyruvate 100 mM (100×) and 1% MEM NEAA (100×)). After washing in hDC medium, the total cell number was determined using the automated cell counting device ViCELL XR Cell Analyzer (Beckman Coulter, Brea, CA, USA) and resuspended in the described medium accordingly. Then, 1.0 × 10^6^/mL hPBMCs in hDC medium were seeded in a 96-well ultra-low attachment plate (Corning, Glendale, CA, USA). LNPs were subsequently added at a dose range of 100 ng, 250 ng, 500 ng, 1000 ng and 2000 ng on top of the cell solution in one quick step and resuspended twice. The cells and LNPs were co-incubated at 37 °C and 5% CO_2_ for 2 h. The transfected hPBMCs were washed, resuspended in fresh hDC medium and further incubated at 37 °C and 5% CO_2_ for 4 h. Extracellular staining was performed after 6 h of total incubation.

### 2.10. In Vitro Cy5 Cell Binding Assay

hPBMCs were seeded at a cell number of 2.0 × 10^5^/mL in hDC medium per well into a 96-well ultra-low attachment plate (Corning, Glendale, CA, USA). LNPs were added in the appropriate dose and co-incubation was performed at 4 °C for 1 h. hPBMCs were washed three times in Flow Buffer (DPBS (1×), 5% FBS (heat inactivated), 5 mM EDTA (500 mM)). Extracellular staining for flow cytometric measurement was performed.

### 2.11. Flow Cytometry 

Monoclonal antibodies for extracellular staining for Thy1.1 transfection assay included CD3-BV421 (BD Biosciences, San Diego, CA, USA; clone: UCHT1), CD4-PE (BioLegend, San Diego, CA, USA; clone: SK3), CD8-APC (BD Biosciences, San Diego, CA, USA; clone: SK1), CD14-BV510 (BD Biosciences, San Diego, CA, USA; clone: MφP9), CD19-PerCP-Cy5.5 (eBiosience, San Diego, CA, USA; clone: SJ25C1) and CD56-PE-Cy7 (BD Biosciences, San Diego, CA, USA; clone: B-159), Thy1.1-BB515 (BD Biosciences, San Diego, CA, USA; clone: OX7). Viability was determined using fixable viability dye eFluor 780 (eBioscience, San Diego, CA, USA). hPBMCs were stained for 20 min at 4 °C in the dark. Cells were washed twice in 150 µL Flow Buffer and the cell pellet was resuspended in 1× Stabilizing Fixative (BD Biosciences, San Diego, CA, USA). Flow cytometric measurements were acquired on a BD FACS Canto II (BD Biosciences, San Diego, CA, USA) and analyzed with FlowJo V10.8.1 software (Tree Star Inc., Ashland, OR, USA). For gating relevant cell groups, cell debris was excluded by side scatter-area (SSC-A) versus forward scatter-area (FSC-A). Singlet cells were gated based on FSC-height (FSC-H) against FSC-area (FSC-A). Thy1.1-expressing cells or Cy5-positive cells were then identified based on their viability. Monocytes were gated as CD14+. B cells and NK cells were gated as CD14−/CD19+ and CD14−/CD56+/CD3−, respectively. CD14−/CD19−/CD56−/CD4+ were identified as CD4+ T cells. CD8+ T cells were gated as CD14−/CD19−/CD56−/CD8+ or CD14−/CD19−/CD56−/CD3+/CD4−.

### 2.12. Statistical Analysis

Data analysis was conducted using GraphPad Prism 9.0 software. The data are presented as mean ± S.D. Comparison of significance between groups was assessed using a two-way analysis of variance (ANOVA II) with Šidák’s multiple comparison correction. * *p* 0.0258, *** *p* < 0.001, **** *p* < 0.0001.

## 3. Results

### 3.1. Particle Size, Zeta Potential and mRNA Incorporation

With our manufacturing protocol, described in the methods and materials part, we were able to obtain particle sizes (Z-Average from dynamic light scattering measurements) within a range of 150–250 nm and a polydispersity index (PDI) ≤ 0.2 (Table 2). LNP formulations manufactured with MC3 overall showed smaller hydrodynamic diameters compared to their DODMA-containing counterparts. Also, the incorporation of pSar-grafted lipids instead of PEGylated lipids led to slightly higher hydrodynamic diameters. The zeta potentials, as measured via electrophoretic light scattering (ELS), were strongly positive at around 30 mV in the glycylglycine buffer (pH 5.7). When measured in DPBS buffer (pH 7.3), the zeta potential decreased to near neutral values due to the ionizable character of the formulations. The fraction of accessible mRNA, determined with the Quant-it™ RiboGreen assay, was low, indicating high encapsulation efficacies.

### 3.2. Fluorescence-Based pK_a_ Determination

The apparent pK_a_ values of mRNA-LNPs were determined using a fluorescence-based TNS assay. The measured intensities in the different pH buffers (covering pH 4.5–9) were fitted with sigmoidal curves, and the inflection points were defined as the apparent pK_a_ of the formulations (Table 3 and Appendix A). All formulations show an apparent pK_a_ of 6.5–6.7, without a notable difference between the varying compositions, in good accordance with former investigations [52].

### 3.3. Small-Angle X-ray Scattering

SAXS was used to investigate the LNP structure for the different compositions. We systematically varied the helper lipid (DOPE vs. DOPC), the ionizable lipid (DODMA vs. MC3), and the stealth moiety (PEG-grafted lipid vs. pSar-grafted lipid) within all LNP formulations. Scattering curves were recorded in the application buffer (containing 10 mM glycylglycine) and in the phosphate buffer at pH 4.5 to mimic the environment in late endosomal uptake processes. The scattering patterns of each formulation in the two different buffers are displayed in Figure 1A.

All curves displayed similar features, in accordance with previous measurements [33] for such systems (Appendix A), dominated by a single broad maximum at around 1 nm^−1^ with no further pronounced patterns, indicative of rather weak lamellar order, while the overall curve shape represented the particle form factor. Here, as data points at very low q could not be detected, no determination of overall size and shape analysis of the particles was possible; however, information on the fractal dimension (surface roughness) of the particles could be derived. We interpreted this peak as resulting from lipid and RNA stacks consisting of very few repeating units and almost no long-range order. Systematic differences depending on the LNP composition and environmental buffer could be discerned.

For quantitative analysis, peak position and width were determined by fitting with Lorentzian functions (formalism see Methods section). Using Bragg’s law, d-spacings were calculated, whereas the model for liquid crystalline order was taken to calculate the correlation length from the peak width (see Methods section, results displayed in Table 4). With its rather small values in the same order of magnitude as the repeat distance, the correlation length was taken for relative comparison between the different systems only. 

The fitting results revealed that the internal organization of the various LNPs sensitively depended on the lipid composition (Figure 1B): MC3-based formulations showed a smaller d-spacing than those with DODMA, and the correlation length was higher in comparison to their DODMA-based analogs, as displayed by direct comparison and mean differences of compared pairs in Figure 1B. DOPE-based formulations showed a smaller d-spacing compared to their DOPC counterparts but to a lower extent as for the ionizable lipid. On the other hand, exchanging the stealth moiety had only a minor effect on d-spacing and correlation length, in accordance with the assumption that the grafted moieties are predominantly present at the particle surface and inserted only to a minor extent into the mRNA/lipid complexes [53,54]. 

For all LNP formulations, we observed a decrease in d-spacing and an increase in correlation lengths when decreasing the pH value of the buffer from 5.7 to 4.5, resulting from increased electrostatic interactions between mRNA and more positively charged ionizable lipids at low pH. These structural changes with decreasing environmental pH were more pronounced for MC3-based LNPs than for DODMA-based analogs.

By using a power law (I~q^−x^, with x as the so-called Porod exponent) to represent the intensity decay of the curves in the form factor describing the low q region, one can gain information on the fractal dimension, or, in other words, the surface roughness, of the LNPs from SAXS measurements [55,56]. A Porod exponent of −4 indicates ideally smooth (flat) interfaces (Porod law), and lower numbers indicate increasing surface roughness. All LNPs in the application buffer had a Porod exponent between −4 and −3.5, indicating compact particles with relatively smooth surfaces. Notably, the Porod exponents were lower for MC3-containing formulations than for those with DODMA, and for LNPs with pSar the values were lower than with PEG LNPs. Interestingly, some pSar-containing formulations (LNP6 and LNP8) showed a further decrease of the Porod exponents when transferred to the pH 4.5 buffer, revealing further increased surface roughness. In contrast, their PEG-lipid-containing counterparts (LNP5 and LNP7) did not show such strong effects. 

To summarize, the SAXS measurements demonstrate that the structure of the mRNA-lipid complexes sensitively depends on the choice of ionizable lipid and the helper lipid. This was even the case when exchanging only the head group of the phospholipid, which was present only in a relatively low fraction in the LNPs. In contrast, the choice of the stealth moiety does not impact the internal structure of the LNP. However, it does influence the surface properties, especially with decreasing the pH value.

### 3.4. In Vitro Transfection Studies

With the information on the internal structure of the LNPs, we investigated mRNA-LNPs on their biological performance by examining transfection and cell binding in primary hPBMCs in vitro. Thy1.1 reporter mRNA was used as cargo to measure transfection efficiency in leukocyte sub populations, such as Monocytes, B-, T- and NK cells. We first evaluated the overall tolerability of all LNP formulations at different doses ranging from 100 ng to 2000 ng. Cell viability for all formulations and dose ranges remained at or above 90%, indicating the good tolerability of the investigated LNPs (Figure 2). Especially Monocytes and to a lesser extent B cells showed Thy1.1 expression and thus a positive transfection. Still, the transfection level in B cells remained under 15% and no transfection signal was detected in T- or NK-cells (Appendix A). 

Monocytes showed a dose-dependent transfection efficiency for all tested LNP compositions with a transfection level up to 95% at the highest dose of 2000 ng (Figure 3). We selected Monocyte transfection as the main focus for further investigations. 

At the highest dose of 2000 ng, we observed different expression levels in all tested LNP formulations. With different ionizable lipids in all LNP formulations (DODMA for LNP1-4; MC3 for LNP5-8), the overall transfection efficiency is higher for MC3-LNPs (Appendix A). For all LNPs, the use of helper lipid DOPE (LNP1-2, LNP5-6) improved the efficiency to a low extent in comparison to DOPC (Appendix A). Interestingly, all pSar-containing LNP formulations (LNP2, LNP4, LNP6, LNP8) showed a significantly higher transfection efficiency than PEG-containing LNP formulations (LNP1, LNP3, LNP5, LNP7), represented in Figure 4. LNP2 reached Thy1.1 expression of greater than 80% in monocytes, while LNP1 only led to 10% Thy1.1 expression. LNP3 showed the lowest transfection rate, with an expression level under 4%, whereas LNP4 reached a Thy1.1 expression above 50%. LNP5 and LNP7 (PEGylated) showed a transfection efficiency of 50%, while LNP6 and LNP8 (pSar-lipid containing) led to a Thy1.1 expression greater than 90% at the same dose (2000 ng). We want to highlight that LNP6 (MC3/DOPE/pSar) achieved the highest Thy1.1 expression in monocytes, starting with more than 75% at the lowest dose (100 ng) and reaching 95%. 

### 3.5. In Vitro Cell-Binding Studies

We correlated the internal lipid structure of the LNPs with their transfection efficiency in vitro and identified that DODMA-based LNPs (LNP1-4) showed a significantly lower monocyte transfection compared to MC3-based LNPs (LNP5-8). We therefore investigated the cellular binding affinity to determine if DODMA-LNPs properly reach monocytes or if the lacking transfection can be explained by disturbed uptake in monocytes. For this, DODMA-based LNPs were co-formulated with an mRNA mix of Thy1.1-encoding mRNA and Cy5-labeled Luc mRNA in a weight ratio of 1:1. pSar-shielded DODMA LNPs (LNP2, LNP4) demonstrated almost 100% of Cy5-positive monocytes compared to PEG-shielded DODMA LNPs (LNP1, LNP3), reaching a cell-binding level between 40–60% (Figure 5A). The choice of helper lipid, again, only minorly improved cell-binding behavior (LNP1 vs. LNP3). Both LNP formulations two and four already reached a cell binding of almost 100%, regardless of the choice of helper lipid. To sum up, pSar as a stealth moiety demonstrated higher cell binding in monocytes compared to PEG-shielded LNPs.

Our in vitro investigations revealed that all tested LNPs exhibited negligible toxicity in all tested dose ranges, while transfection efficiency and cell-binding behavior varied depending on the LNP composition. LNPs containing MC3 as an ionizable lipid, DOPE as a helper lipid, and pSar-grafted lipid showed improved transfection rates among all formulations and exhibited the highest potency in vitro, with high efficacy even at a low tested dose range. Moreover, pSar-containing DODMA-LNP formulations showed outstanding cell-binding affinity to monocytes. These findings substantiate that the choice of specific lipid components, especially regarding the stealth moiety, plays a critical role in enhancing the transfection efficiency of LNPs on primary hPBMCs in vitro. 

## 4. Discussion

Lipid nanoparticles as non-viral RNA delivery vehicles have great potential in various applications, such as vaccines against infectious diseases [4,8,9], protein-replacement therapies [2] or novel cancer immunotherapy approaches [4,18]. Currently approved LNP formulations are characterized by a very specific lipid composition, consisting of four primary components: ionizable-cationic lipid, cholesterol, helper lipid and a PEG stealth moiety [10,57]. The application of mRNA in other therapeutic settings as mentioned above will require the development of delivery systems optimized for the respective application. So far, the development of such tailored therapeutics is challenged by the limited understanding of the intramolecular structure–function features in LNPs, especially regarding the interplay between ionizable lipids, helper lipids and stealth moieties. Therefore, here we have investigated the influence of selected lipids on the structural and functional characteristics of the LNPs. By using SAXS together with other techniques for physicochemical characterization and transfection studies, we were able to directly determine the way the tested molecular groups affected internal LNP structure and also biological functionality.

For the MC3-LNPs, which were more active than the ones with DODMA, the repeat distance was smaller and the long-range order higher, although, with its two linoleyl moieties, MC3 is bulkier and spans a larger distance. 

This we interpreted as due to the stronger electrostatic interactions between the charged moieties of the RNA and the lipid head group. Accordingly, the effect increased at low pH, when the charge density at the ionizable lipid head group was higher. Notably, the pK_a_ values, which are frequently taken as indicators for efficacy, were rather similar for DODMA and MC3. Therefore, the information on structural and electrostatic coherencies obtained here may provide a supplementary indication to explain functional aspects of the ionizable lipids. MC3 facilitates endosomal escape [58] and shows great potential for RNA delivery [16,59]. Our in vitro data can confirm the improved transfection capability of MC3 in the tested cells and provide a potential structural foundation for these findings.

Also, the choice of the helper lipid, even only present at 10 mol% in the lipid mixture, resulted in measurable effects of structure and activity. With DOPE as a helper lipid, the d-spacing was lower than with DOPC, and the activity of the LNPs was higher. For the effect on the structure, it is plausible to assume that the bulkier PC headgroup in comparison to PE contributes to a higher repeating distance. Regarding activity, more complex cooperative effects on the membrane organization may play a role. The fusogenic properties of PE lipids may have facilitated uptake and endosomal release. It has been reported that the helper lipid in LNPs is enriched in the shell of the particles [54]; therefore, even at the low molar fraction as present here, the type of head group can affect membrane interactions on endosomal uptake and release. As well, there may be a specific preference of PE-containing nanoparticles to immune cells as tested here. In fact, for another type of lipid nanoparticles (lipoplexes), DOPE has been identified as the most suitable lipid for targeting antigen-presenting cells in vivo [18,19]. Notably, in these experiments, the presence of PE leads as well to very low expression of the lipoplexes in the liver. Therefore, for future developments, LNPs comprising PE could be of particular interest for extrahepatic targeting of the RNA, which is still an unmet need for many applications. The currently established LNPs (including those used for the COVID-19 vaccines), which comprise PC as the phospholipid moiety [10], are known to target the liver to a large extent. 

Concerning the polymer-grafted lipids, we found further evidence for the potency of pSar as a stealth moiety for engineering particles with an improved efficacy profile. There is a need for alternatives to PEG-containing LNPs, which still represent the gold standard for RNA delivery due to their adverse side-effects caused by anti-PEG antibodies [60,61]. pSar demonstrates comparable stealth-like properties and is a bio-based and biodegradable material [39,62]. In previous studies [33], we showed that pSar as a stealth component in mRNA LNPs provides particles with comparable physicochemical properties as LNPs manufactured with PEG moieties together with lower toxicity and improved protein secretion in HepG2 cell line in vitro as well as in Balb/C mice in vivo. It is hypothesized that pSar used in lipid delivery systems can circumvent the ABC phenomena based on multiple injection experiments in rats in comparison to PEG-grafted formulations [63]. The potential of pSar to be readily functionalized brings further possibilities for the targeting of delivery systems. End-group modifications with different amino acids as well as antibodies or fragments of them allows the selection of charge and molecular properties of the particle surface and therefore control of circulation and targeting properties [64]. Careful SAXS data analysis allowed us to correlate distinct structural properties of the pSar LNPs with the improved transfection results. The elevated surface roughness of the pSar-LNPs compared to those with PEG, which even increased at low pH, may foster interaction with the oppositely charged endosomal membrane, resulting in the facilitated rupture of the membrane and the release of the mRNA into the cytosol [65,66]. The at least partial positive charge of the amine head group of pSar may further contribute to this effect. Combining MC3, with its good RNA transfection properties, with DOPE as a helper lipid and pSar as a surface modification, we generated an LNP composition with optimal conditions for transfecting hPBMCs. Thus, we have determined certain compositional and structural ‘fingerprints’ of LNPs which led to improved transfection efficacy, here demonstrated for immune cells. Systems with higher activity were obtained when the internal order and packing density were higher and with increased fractal dimension of the particles. Further experiments to investigate targeting and efficacy in vivo in correlation with the structural observations will be necessary to fully elucidate these structure–function correlations. This insight can provide useful guidance for the organization of development experiments toward optimized LNPs for future applications.

## 5. Conclusions

In this study, the influence of discrete lipid characteristics on the structure and functionality of LNP formulations was determined. SAXS analysis provided valuable insights into the overall morphology and internal structure of the LNPs, while activity was tested on hPBMCs in vitro. Gaining a deeper understanding of these relationships can prove highly valuable for the development of safe and efficient delivery systems and the implementation of quality control measures. pSar-lipids, as an alternative to PEG-lipids, were further validated for the assembly of LNP formulations with controlled size and improved transfection efficiency. The findings may serve as a basis to derive general rules for the development of tailored LNP formulations.

## Figures and Tables

**Figure 1 pharmaceutics-15-02068-f001:**
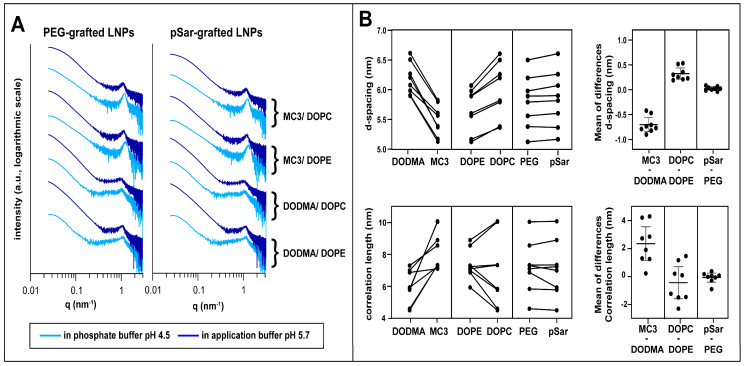
Small-angle X-ray scattering (SAXS) investigation. (**A**) SAXS patterns of different LNP formulations in phosphate buffer (pH 4.5, light blue) and application buffer (pH 5.7, dark blue). Formulations are displayed according to their composition with PEG-grafted LNPs (left) and pSar-grafted LNPs (right) and their ionizable and helper lipid are displayed right to the scattering patterns. Scattering patterns are vertically shifted for better visualization. (**B**) Comparison between LNP formulations in d-spacing (top). Investigated pairs generated with formulations only differing in one lipid component (ionizable lipid, helper lipid, stealth lipid). Mean of the differences in the compared formulations is shown on the right and represents the mean factor in which the formulations differ in d-spacing when comparing the investigated pairs; the same procedure for correlation length is at the bottom. Data displayed as mean ± S.D.

**Figure 2 pharmaceutics-15-02068-f002:**
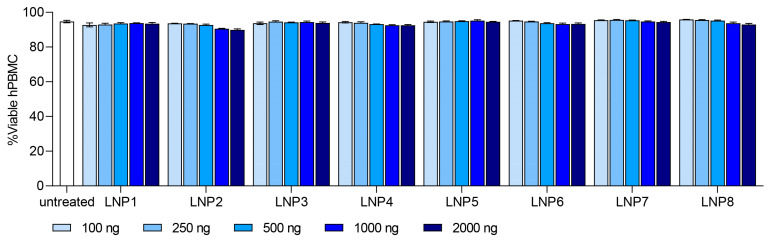
In vitro tolerability of LNP formulation 1–8 in human peripheral blood mononuclear cells (hPBMC). Dose ranged from 100 ng to 2000 ng. Viability of each LNP formulation is shown as %Viable hPBMC. Data are presented as mean ± S.D., *n* = 3 technical replicates per LNP formulation.

**Figure 3 pharmaceutics-15-02068-f003:**
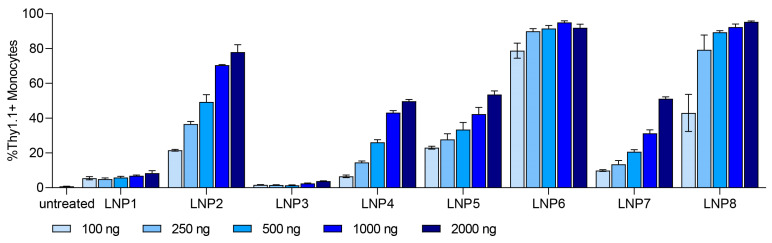
In vitro dose-dependent transfection efficiency of Monocytes for LNP formulations 1–8. Dose ranged from 100 ng to 2000 ng. Transfection efficiency of each LNP formulation is shown as %Thy1.1+ Monocytes. Data are presented as mean ± S.D., *n* = 3 technical replicates per LNP formulation.

**Figure 4 pharmaceutics-15-02068-f004:**
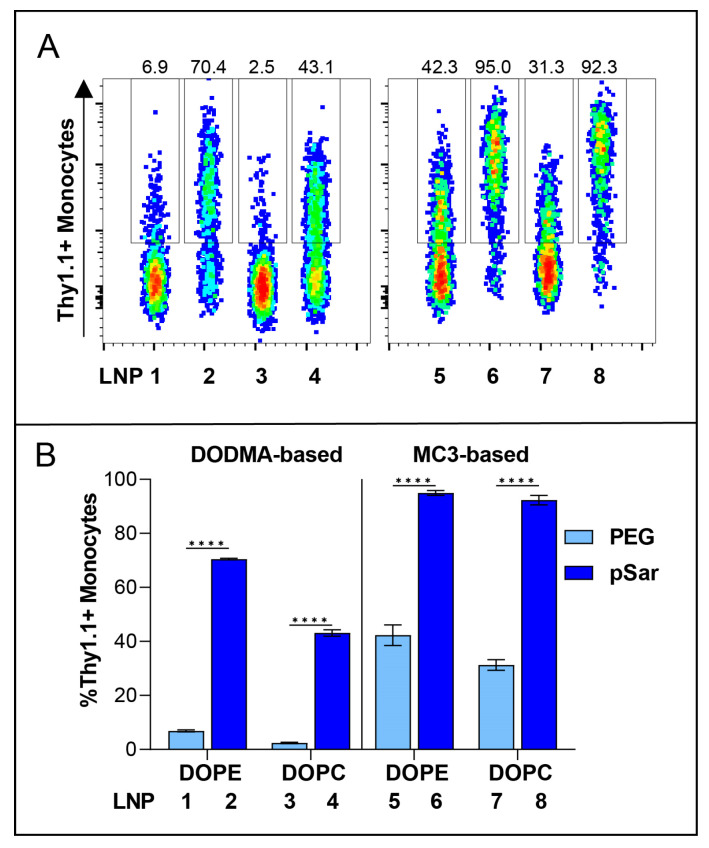
In vitro transfection efficiency of Thy1.1 RNA containing LNPs at a dose of 1000 ng in hPBMC, Monocytes as representative cell group. (**A**) Thy1.1-expressing Monocytes analyzed by flow cytometry. Numbers indicate the percentage of Thy1.1+ Monocytes. (**B**) Transfection efficiency of all PEG-lipid versus pSar-lipid LNPs shown as %Thy1.1+ Monocytes. Data are presented as mean ± S.D., analyzed by a two-way ANOVA with Šidák’s multiple comparison test, **** *p* < 0.0001, *n* = 3 technical replicates per LNP formulation.

**Figure 5 pharmaceutics-15-02068-f005:**
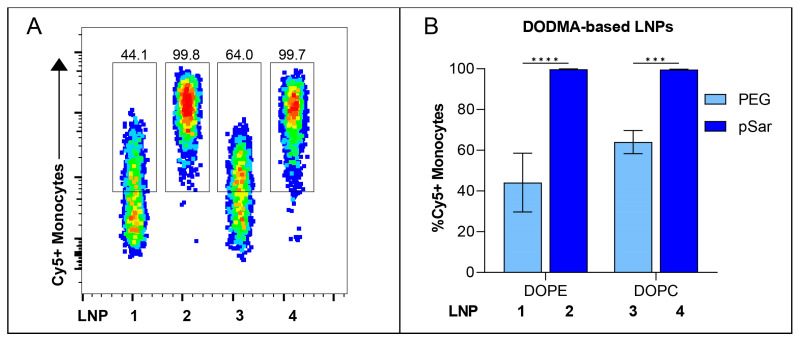
In vitro cell-binding studies of Cy5-labeled RNA containing DODMA-LNP at a dose of 1000 ng in hPBMCs. (**A**) Cy5-labeled RNA-positive Monocytes analyzed by flow cytometry. Numbers indicate the percentage of Cy5+ Monocytes. (**B**) Cell-binding efficiency of each DODMA-LNP formulation is shown as %Cy5+ Monocytes. Data are presented as mean ± S.D., analyzed by a two-way ANOVA with Šidák’s multiple comparison test, *** *p* < 0.001, **** *p* < 0.0001, *n* = 3 technical replicates per LNP formulation.

**Table 1 pharmaceutics-15-02068-t001:** Composition of investigated formulations.

Formulation	Helper Lipid	Helper Lipid (mol%)	Ionizable Lipid	Ionizable Lipid (mol%)	Cholesterol (mol%)	Stealth Moiety	Stealth Moiety (mol%)	mRNA (mol%)
LNP 1	DOPE	10	DODMA	40	48	PEG	2	8
LNP 2	DOPE	10	DODMA	40	48	pSar	2	8
LNP 3	DOPC	10	DODMA	40	48	PEG	2	8
LNP 4	DOPC	10	DODMA	40	48	pSar	2	8
LNP 5	DOPE	10	MC3	40	48	PEG	2	8
LNP 6	DOPE	10	MC3	40	48	pSar	2	8
LNP 7	DOPC	10	MC3	40	48	PEG	2	8
LNP 8	DOPC	10	MC3	40	48	pSar	2	8

All formulations were manufactured with an N/P ratio of 5. Sample names for lipid nanoparticle (LNP) formulations will be used throughout the manuscript: 1,2-dioleyloxy-3-dimethylaminopropane (DODMA); DLin-MC3-DMA (MC3); 1,2-dioleoyl-sn-glycero-3-ethanolamine (DOPE), 2-Dioleoyl-sn-glycero-3-phosphocholine (DOPC), C16-PEG2000-Ceramide (PEG); polysarcosine BA12-50 (pSar).

**Table 2 pharmaceutics-15-02068-t002:** Physicochemical characterization of mRNA LNPs in application buffer (10 mM glycylglycine, pH 5.7). Data presented as mean ± S.D., *n* = 3.

Formulation	Composition	Diameter (nm)	PDI	Zeta Potential (mV)	Inaccessible mRNA (%)
**LNP 1**	**DODMA/DOPE/PEG**	201 ± 15	0.202 ± 0.043	27 ± 7	95 ± 3
**LNP 2**	**DODMA/DOPE/pSar**	239 ± 21	0.173 ± 0.028	34 ± 6	91 ± 4
**LNP 3**	**DODMA/DOPC/PEG**	221 ± 15	0.205 ± 0.021	28 ± 5	92 ± 4
**LNP 4**	**DODMA/DOPC/pSar**	242 ± 33	0.193 ± 0.040	35 ± 3	90 ± 4
**LNP 5**	**MC3/DOPE/PEG**	174 ± 07	0.124 ± 0.021	30 ± 3	96 ± 2
**LNP 6**	**MC3/DOPE/pSar**	196 ±13	0.170 ± 0.015	33 ± 2	93 ± 2
**LNP 7**	**MC3/DOPC/PEG**	159 ± 15	0.117 ±.0.024	27 ± 6	97 ± 1
**LNP 8**	**MC3/DOPC/pSar**	197 ± 17	0.156 ± 0.028	34 ± 1	94 ± 2

**Table 3 pharmaceutics-15-02068-t003:** Apparent pK_a_ of all LNP formulations via fluorescence-based 2-(p-toluidino)-6- naphthalene sulfonic acid (TNS) assay. Data presented as mean ± S.D., *n* = 3.

Formulation (DODMA)	Apparent pK_a_	Formulation (MC3)	Apparent pK_a_
**LNP 1**	6.5 ± 0.0	**LNP 5**	6.7 ± 0.0
**LNP 2**	6.5 ± 0.1	**LNP 6**	6.6 ± 0.0
**LNP 3**	6.5 ± 0.2	**LNP 7**	6.5 ± 0.1
**LNP 4**	6.6 ± 0.1	**LNP 8**	6.5 ± 0.1

**Table 4 pharmaceutics-15-02068-t004:** Results of SAXS data analysis for the LNP formulations as applied and in phosphate buffer pH 4.5, respectively.

Formulation	Application Buffer (pH 5.7)			Phosphate Buffer (pH 4.5)		
	d-Spacing (nm)	Correlation Length (nm)	Porod Exponent	d-Spacing (nm)	Correlation Length (nm)	Porod Exponent
**LNP 1**	6.0 ± 0.1	6.9 ± 0.3	−3.9 ± 0.1	5.9 ± 0.1	7.3 ± 0.2	−3.4 ± 0.1
**LNP 2**	6.1 ± 0.1	4.6 ± 0.3	−3.9 ± 0.1	5.9 ± 0.1	7.0 ± 0.2	−3.4 ± 0.1
**LNP 3**	6.5 ± 0.1	5.9 ± 0.2	−3.9 ± 0.1	6.2 ± 0.1	5.8 ± 0.1	−3.9 ± 0.1
**LNP 4**	6.6 ± 0.1	4.5 ± 0.2	−3.9 ± 0.1	6.3 ± 0.1	5.8 ± 0.2	−3.9 ± 0.1
**LNP 5**	5.6 ± 0.1	7.1 ± 0.2	−3.8 ± 0.1	5.1 ± 0.1	8.6 ± 0.2	−3.3 ± 0.1
**LNP 6**	5.8 ± 0.1	7.3 ± 0.2	−3.5 ± 0.1	5.2 ± 0.1	8.9 ± 0.2	−2.6 ± 0.1
**LNP 7**	5.6 ± 0.1	7.2 ± 0.3	−3.7 ± 0.1	5.4 ± 0.1	10.0 ± 0.1	−3.5 ± 0.1
**LNP 8**	5.8 ± 0.1	7.3 ± 0.3	−3.6 ± 0.1	5.4 ± 0.1	10.1 ± 0.2	−2.7 ± 0.1

## Data Availability

The data presented in this study are available on request from the corresponding author.

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
