# Peer review of "Polysarcosine-Functionalized mRNA Lipid Nanoparticles Tailored for Immunotherapy"

_pharmaceutics, 2023, doi:10.3390/pharmaceutics15082068_

Round 1
Reviewer 1 Report
The manuscript entitled "Polysarcosine-functionalized mRNA lipid nanoparticles tailored for immunotherapy" is an original research investigating a very interesting and attractive scientific topic. The mRNA delivery is gaining increasing scientific role in the prevention and therapy of social burdening diseases. The application of various lipids and their relation to the function of the prepared nanoparticles can allow diversity in the application for mRNA delivery. The authors in the current manuscript perform a systematic, scientifically sound and logical investigation on the influence of the lipid composition on the efficacy and structure of the lipid nanoparticles. The manuscript is overall well organized and fluently written. It requires minor technical corrections prior to being published (e.g. p.5 line 227 - there are two points; p. 10 line 383 - there is ")," which is inappropriate; p. 14 line 539 - a verb is missing "which still an unmet need..."). Otherwise, I recommend it being published in its current form.
Reviewer 2 Report
The authors describe in detail the formulation and function of the mRNA lipid nanoparticles system based on SAXS analysis of the particulate structure. These results will be helpful and informative for researchers in the field of biomaterials. The reviewer thinks the authors’ study in this manuscript is quite interesting, suggestive, and well-organized.
From these considerations, the reviewer recommends accepting for publication in “Pharmaceutics,” if the following issue is resolved.
1) A "Conclusion" should be added that clearly describes the significance and importance of the results obtained by the authors.
Reviewer 3 Report
Prior to acceptance, some points need to be addressed:
1. The health hazards caused by PEG-conjugated lipids have been cited by the authors as a major impetus for the present study. While this is indeed a basic problem affecting the therapeutic efficacy of LNPs, the toxic properties of cationic lipids as another important ingredient should be mentioned as well.
For example, this topic has been recently covered by Segalla (2023) Chemical-physical Criticality and Toxicological Potential of Lipid Nanomaterials Contained in a COVID-19 mRNA Vaccine. International Journal of Vaccine Theory, Practice, and Research 3(1): 787-817. Concerning the cationic and PEGylated lipid components of one of the major COVID-19 vaccines, the following statement is quoted, highlighting the fact that more research on their effects on human health is warranted:
„In addition to being unknown to the European Pharmacopoeia, the two lipid components ALC-0315 and ALC-0159 are not even reported in the C&L inventory [Classification Labelling Inventory of the European Chemicals Agency; ECHA]. Consequently, they do not have a REACH registration number and their CLP classification is not known. In other words, their general toxicological profile is not officially known — neither as substances, nor as nanoforms made up of them.“
2. The authors mention in Methods, l. 297 that cell debris was excluded from flow cytometry. Were all kinds of cell debris fully quantified by the viability assay?
3. In the diagrams depicted in both Figures S3 and S4, the labels of the LNP samples were erroneously confused:
In Figure S3, the LNP pairs should be 3-7 and 2-6 – instead of 2-6 and 3-7 as labeled.
In Figure S4, the LNP pairs should be 1-3, 5-7, 2-4, and 6-8 – instead of 1-2, 5-6, 3-4, and 7-8 as labeled.
4. The authors mention the „successful global Covid-19 vaccination campaign“ in a prominent position at the end of the first sentence of the Abstract. As an assessment of this vaccination campaign is not important for the present study, such a statement is rather dispensable until a final evaluation of the benefits vs. near- and long-term adverse effects and their costs will be obtained during the next years.
In fact, substantial evidence has been accumulating since 2021 in hundreds of scientific publications in peer-reviewed journals as well as official information from government authorities of several states, which altogether seriously questions the benefit of the Covid-19 vaccination campaign, suggesting the need for more research and collection of data. A few important examples are subsequently listed
4a. For Western Australia, the official report for 2021 states a total AEFI (adverse events following immunisation) rate for COVID-19 vaccines of 264.1
per 100,000 doses, while that for all other vaccines were 11.1 per 100,000 doses. https://www.health.wa.gov.au/~/media/Corp/Documents/Health-for/Immunisation/Western-Australia-Vaccine-Safety-Surveillance-Annual-Report-2021.pdf
4b. The excess mortality in German during 2020-2022 has been statistically investigated by Kuhbandner and Reitzner (2023) Estimation of Excess Mortality in Germany During 2020-2022. 2023 May 23; Cureus 15(5):e39371. The final paragraph of their Conclusions section is quoted as: "As a starting point for further investigations explaining these mortality patterns, we compared the excess mortality to the number of reported COVID-19 deaths and the number of COVID-19 vaccinations. This leads to several open questions, the most important being the covariation between the excess mortality, the number of COVID-19 deaths, and the COVID-19 vaccinations.".“
4c. Josh Guetzkow and Retsef Levi recently criticized insufficient transparency concerning COVID-19 vaccine clinical trials, which is particularly relevant for data on the manufacturing process, as published in a letter to the editor in the British Medical Journal (www.bmj.com/content/378/bmj.o1731/rr-2): Rapid Response: Effect of mRNA Vaccine Manufacturing Processes on Efficacy and Safety Still an Open Question. Their major point is that due to differences in production processes, it cannot be assumed that the COVID-19 vaccines, whose "effectiveness" has been tested in clinical trials and is the basis of all approvals, are in any way comparable in effectiveness to COVID-19 vaccinations that have been mass-produced using a different process. This conclusion is based on the data in publicly released documents from Pfizer (www.phpt.org, as consequence of a Freedom of Information Act (FOIA) request).C
There are essentially no language issues.
Reviewer 4 Report
In this manuscript, the authors used SAXS as a powerful tool to determine the influence of the respective lipids on the LNP structure. However, the data presented in current version are insufficient. Some specific comments are included below.
1: What would be the advantages of polysarcosine-functionalized LNPs in the aspect of stability and targeting ability?
2: It is necessary to recheck the significant digit in Table 2.
3: Further animal studies in correlation with the immunotherapy and immunogenicity should be investigated.
4: It would be more rigorous if the endosomal escape study is included in vitro experiments.
Acceptable but could be further improved.
